# Application of ZnO-Based Nanocomposites for Vaccines and Cancer Immunotherapy

**DOI:** 10.3390/pharmaceutics11100493

**Published:** 2019-09-26

**Authors:** Prashant Sharma, Na-Yoon Jang, Jae-Won Lee, Bum Chul Park, Young Keun Kim, Nam-Hyuk Cho

**Affiliations:** 1Department of Microbiology and Immunology, Seoul National University College of Medicine, Seoul 03080, Korea; psharma@snu.ac.kr (P.S.); zcbtnja@snu.ac.kr (N.-Y.J.); zaenner@snu.ac.kr (J.-W.L.); 2Department of Biomedical Sciences, Seoul National University College of Medicine, Seoul 03080, Korea; 3Department of Materials Science and Engineering, Korea University, Seoul 02841, Korea; p23rd@korea.ac.kr; 4Research Institute of Engineering and Technology, Korea University, Seoul 02481, Korea; 5Institute of Endemic Disease, Seoul National University Medical Research Center and Bundang Hospital, Seoul 03080, Korea

**Keywords:** ZnO nanocomposite, vaccine, cancer, immunotherapy, toxicity, immune cells

## Abstract

Engineering and application of nanomaterials have recently helped advance various biomedical fields. Zinc oxide (ZnO)-based nanocomposites have become one of the most promising candidates for biomedical applications due to their biocompatibility, unique physicochemical properties, and cost-effective mass production. In addition, recent advances in nano-engineering technologies enable the generation of ZnO nanocomposites with unique three-dimensional structures and surface characteristics that are optimally designed for in vivo applications. Here, we review recent advances in the application of diverse ZnO nanocomposites, with an especial focus on their development as vaccine adjuvant and cancer immunotherapeutics, as well as their intrinsic properties interacting with the immune system and potential toxic effect in vivo. Finally, we summarize promising proof-of-concept applications as prophylactic and therapeutic vaccines against infections and cancers. Understanding the nano-bio interfaces between ZnO-based nanocomposites and the immune system, together with bio-effective design of the nanomaterial using nano-architectonic technology, may open new avenues in expanding the biomedical application of ZnO nanocomposites as a novel vaccine platform.

## 1. Introduction

The development of vaccines against various infections is one of the most significant achievements of modern medical sciences [1]. Immunotherapeutic vaccines have also been developed in the last few decades to control cancers and other non-infectious diseases [2,3,4]. However, there have been many challenges and limitations that have hampered the development of potent vaccines against diverse emerging and re-emerging diseases, such as antigen selection, proper antigen delivery, and adjuvant engineering [5]. Nanotechnology has been exploited to overcome these issues, rapidly advancing the design and development of vaccines [2,6,7,8,9,10]. Diverse nanocomposites in various formulations and compositions have been shown to improve the efficiency of antigen delivery and immunogenicity by augmenting antigen processing, enhancing antigen stability, and sustaining release. Nanocomposites applied for the development of diverse vaccines include those made from organic polymers such as poly(lactic-*co*-glycolic acid) (PLGA), poly(l-lactic acid) (PLLA), polyethylene glycol (PEG), alginate, inulin, and chitosan, as well as inorganic nanoparticles (NPs) such as silica-based NPs [10,11].

Metal-based inorganic nanocomposites, including zinc oxide (ZnO), titanium dioxide, and iron oxide, have been recently applied as vaccine carriers due to their rigid structures, long shelf life, and the capability to tailor their intrinsic adjuvant-like properties and immunomodulatory functions [12]. ZnO, a well-documented Food and Drug Administration (FDA)-approved material, has been widely used for several biomedical applications due to its biocompatibility, stability, and cost effectiveness [13]. Furthermore, elemental Zn affects multiple aspects of the immune system and can be efficiently excreted from the body through various routes, including sweat, urine, and feces, thereby reducing the chances of accumulation within the body [14,15,16]. Nevertheless, ZnO nanocomposites seem to cause a degree of in vivo toxicity and inflammatory responses. ZnO NPs have been shown to induce lung inflammation by the production of pro-inflammatory cytokines in a myeloid differentiation primary response protein-88 (MyD88)-dependent manner via Toll-like receptor (TLR) signaling pathways when instilled into the respiratory tract [17]. When subcutaneously or orally administered, ZnO nanocomposites have also been shown to cause inflammatory reactions [18,19,20] by Zn^2+^ dissolution and reactive oxygen species (ROS) generation [21].

Given the immune-modulatory effect of ZnO, it has also been applied as an adjuvant system for various vaccines. Although not fully characterized, a few reports have shown the use of ZnO nanocomposites as a promising immune modulator when combined with vaccine antigens [7,22,23,24,25,26]. Additionally, various forms of ZnO nanocomposites have also been shown to enhance anti-cancer immunity when used as tumor antigen carriers [8,26].

### 1.1. Synthesis of ZnO Nanostructure

Various methods have been introduced to synthesize ZnO nanostructures and are summarized well in previous reviews [27,28]. The size, shape, composition, and surface property of the ZnO nanostructures can be finely controlled according to experimental conditions, such as chemical reactants, solvents, temperature, and time [28]. Since ZnO nanocomposites have been reported to react with in vivo systems differentially, depending on its size, shape, and surface property, it is necessary to select an appropriate synthesis method according to the desired purposes. For spherical ZnO NPs, precipitation, micro-emulsion, and sol–gel methods are mainly used [27]. In particular, the sol–gel technique has been the most popular method ever since it was first reported by Spanhel and Meulenkamp [29,30]. In this method, ZnO NP is obtained from zinc hydroxide, synthesized by reacting Zn acetate dihydrate with hydroxyl anion (OH^−^) in ethanol. Hydrothermal method is typically used to synthesize ZnO nanowires (NWs) [28]. ZnO NWs can be grown on specific substrates using ZnO NPs as seeds. Heterogeneous nucleation on ZnO NP seeds is thermodynamically favorable when compared to homogeneous nucleation in a reacting solution. Since the surface energy of ZnO varies depending on the crystal plane, the aspect ratio of ZnO nanocomposites can be controlled by facet-controlled growth using various surfactants [28,31,32]. In addition, ZnO can be synthesized in various forms by controlling pH, temperature, and Zn precursors [28]. The hydrothermal method can be also applied for generating ZnO hybrid structures by using various organic and inorganic substrates as templates because ZnO NWs can be grown regardless of substrate type [26,33]. Green biosynthesis using nontoxic chemicals has been developed recently to prepare biocompatible ZnO nanostructures [34]. It is a promising non-toxic and biocompatible method because it utilizes biological products obtained from plants, fungi, food, algae and bacteria.

### 1.2. History of ZnO Application in Biomedical Fields

ZnO is produced by heating zinc ore in a shaft furnace. Metallic zinc is liberated as a vapor, which ascends a flue and condenses as an oxidized form for various applications [35,36,37]. ZnO has long been used in our daily life as ingredient in medicine, cosmetics, and food additives. Piezoelectric ZnO has also been widely applied in optoelectronics, sensors, transducers, and energy conversion. Historical and modern evidence suggest that ZnO has low toxicity and biodegradability [13], with potential applications in biomedical sciences [35,38]. The Charaka Samhita, an ancient Indian medical textbook written around 500 B.C., mentions a probable use of ZnO, called Pushpanjan in the text, for eyes and open wound treatment [39]. A Greek physician, Dioscorides, introduces ZnO ointment in the first century A.D. [36]. Avicenna, a Persian philosopher and physician, also discussed ZnO in his book, The Canon of Medicine, as a preferred cure for multiple skin diseases, including skin cancer [40].

Material science blossomed in the 20th century, and ZnO was one of the first metal oxide materials to be investigated in detail. ZnO is a semiconductor material with a direct wide bandgap energy (3.37 eV) and a large excitation binding energy (60 meV) at room temperature [41]. Its unique semiconductor properties make it appropriate for various photo-induced applications including photocatalyst and photoemission [42]. When ZnO absorbs light in the ultra-violet (UV) range shorter than 400 nm, electron–hole pairs can be generated in the conduction and valence bands. The conduction band is the band of electron orbitals that excited electrons (e^−^) jump into from valence band. The valence band is the outermost electron orbital of an atom of a material. The bandgap is the energy difference between the highest occupied energy state of the valence band and the lowest unoccupied state of the conduction band. Excited electrons are promoted to the conduction band, leaving holes (h^+^) in the valence band and both electrons and holes can be involved in the physiological activity of cells. They can generate ROS such as superoxide and hydroxyl radicals by reaction with oxygen or water [27]. Inversely, recombination of electron–hole pair leads to emission of a photon, which allows ZnO to be used for bio-imaging [43]. Among the various metal oxides with wide bandgap, ZnO is in the spotlight for its low-cost fabrication, various structure, and facile surface modification [27,41]. ZnO crystallizes into either a cubic zinc-blende or hexagonal wurtzite structure in which each anion is surrounded by four cations at the corners of a tetrahedron, and vice versa, respectively. The crystal structures shared by ZnO are wurtzite, zinc blende, and rock salt. The hexagonal wurtzite is the most thermodynamically stable phase [41]. Electron diffraction data on ZnO was first presented in 1935 [44]. Due to their unique physical and chemical properties, ZnO-based nanocomposites have shown promising potential in bio-imaging and drug delivery during the last decade [7,45,46]. Recent advance in nano-engineering technologies to generate ZnO composites with unique three-dimensional structures and surface characteristics enable them to adjust optimally designed artificial tools for various in vitro and in vivo applications (Figure 1). ZnO NPs chemically conjugated with acid-sensitive polymers were shown to be taken up by cancer cells through endocytosis, and release a preloaded anti-cancer drug into cellular lysosomes in a pH-dependent manner [47]. In addition to chemical drugs and protein antigens, plasmid DNA can also be delivered and expressed in eukaryotic cells when associated with ZnO nanostructures [7,8,48]. Silica-coated tetrapod-like ZnO nanostructures bind plasmid DNA through electrostatic interactions and mediate intracellular delivery for expression of the associated DNA in mammalian cells [46,49]. Furthermore, fabrication of a unique three-dimensional (3D) structure of ZnO tetrapod showed antiviral activity against human herpesviruses (HSV) [50,51]. We also reported that an iron oxide–ZnO core-shell NP can efficiently deliver a tumor antigen fused with ZnO-binding peptide (ZBP) into dendritic cells (DCs) and potently enhance antigen-specific immunity in vivo when applied for cancer immunotherapy [8]. Immunization of tumor-bearing mice with radially grown ZnO on PLLA microfibers can significantly change the tumor microenvironment, systemic immune responses and inhibit tumor growth in vivo [26]. Therefore, application of ZnO nanocomposites as an adjuvant system for vaccine development is now attracting more attention in related biomedical fields.

In this review, we discuss nano-bio interfaces of ZnO nanocomposites in various forms and their potential as immune adjuvant system for vaccines and cancer immunotherapy. In addition, the underlying mechanisms responsible for the intracellular delivery of associated antigens and the immune stimulatory function of ZnO nanocomposites are described in detail.

## 2. Interactions of ZnO Particles with Immune Cells

ZnO particles (~5 μm in size) are readily ingested by a variety of phagocytes such as monocytes, macrophages, and DCs, and then exert the immune-modulatory effects on these innate immune cells [56,57,58]. In general, these stimulatory effects are mediated by intracellular Zn^2+^ dissolution in the lysosomes and generation of ROS, resulting in the release of inflammatory cytokines and cellular activation of immune cells. These effects result in antigen-specific adaptive immune responses when co-administered with an antigen, as well as cellular toxicity in vivo [24,57,59]. In addition, non-immune cells, such as epithelial cells and neuronal cells, are affected by ZnO NPs, primarily due to Zn^2+^ dissolution and ROS generation [60,61,62,63]. Moreover, it has been shown that cancer cells are more susceptible to ZnO NP exposure than normal cells [64]. We primarily focus on the interaction of ZnO nanocomposites with innate phagocytic cells and cancer cells in this review.

### 2.1. ZnO Uptake by Immune Cells

ZnO NPs are readily phagocytosed due to their peculiar electrostatic characteristics (Figure 2) [27]. ZnO exhibits a unique surface charge behavior in which a neutral hydroxyl group (isoelectric point: pH 9–10) on its surface shifts its surface charge depending on the surrounding pH [65]. At high pH, ZnO transfers its protons from its surface to its surroundings to become ZnO^−^; at low pH, ZnO gains protons on its surface from the surrounding aqueous environment to become ZnOH^2+^. Hence in physiological conditions (lower than pH 7.4), ZnO generally exists as ZnOH^2+^. This is critical as negatively charged cell membranes attract the positively charged ZnO NPs to enable intracellular delivery of particles into cells [27]. The uptake of ZnO NPs can occur as early as 1 h post incubation with phagocytes [8]. ZnO NPs are then transported to the lysosomes whereupon Zn^2+^ dissolves and ROS is generated [21]. Of note, the majority of ZnO particles (~μm in diameters) are taken up by macrophages via scavenger receptor, Fcγ receptor (FcγR), and complement receptor, while ZnO NPs are cleared by clathrin or caveolae mediated endocytosis, suggesting that the route of cellular uptake depends on size [57].

### 2.2. Stimulation of Toll-Like Receptors (TLRs)

Induction of innate immune responses by ZnO nanocomposites might be mediated by direct recognition by TLRs (TLR4 or TLR6) (Figure 3) [17,66]. However, the mechanism of ZnO nanocomposite interactions with TLRs is ambiguous. One proposed hypothesis is that when NPs are introduced in vivo, various proteins such as albumin adsorb onto the surface of the NPs, forming a protein corona [67]. The adsorbed proteins may include immune-stimulatory proteins and/or other stimulatory molecules, which interact with the TLRs. Alternatively, the nano-sized NPs themselves might mimic TLR ligands and bind directly to TLRs. Nonetheless, the expression of various TLRs such as TLR 1, 2, 4, and 6, as well as downstream signaling molecules, including MyD88, IL-1 receptor associated kinase 1 (IRAK-1), and TNFR-associated factor 6 (TRAF6), significantly increase in macrophages when ZnO NPs are introduced with an antigen in vivo [22]. Enhanced expression of these molecules further activates NF-κB signaling, and thereby induces the release of pro-inflammatory cytokines and activates innate phagocytes (Figure 3).

### 2.3. Zn^2+^ Dissolution and ROS Generation

Once internalized, ZnO NPs continuously dissociate into Zn^2+^ ions in acidic lysosomes [21]. ZnO NPs, when compared to TiO_2_, CeO_2_, and Al_2_O_3_ NPs, preferentially resulted in the highest increase in phosphorylation of p38, extracellular signal-regulated kinase (ERK), and c-Jun N-terminal kinase (JNK) of mitogen-activated protein kinase (MAPK) pathways, in the polyhydroxyalkanoate (PHA)-activated Jurkat cells, a T cell line [68]. Activation of MAPK pathways was also observed in other immune cells [66,69]. Additionally, ZnO NPs induced the degradation of IκBα, the NFκB inhibitor, to enhance NFκB signaling [68]. Zn^2+^ dissolution is thought to be the underlying mechanism for these effects as equal molar treatment with ZnCl_2_ showed similar results in a MAPK pathway-dependent manner. Taken together, Zn^2+^ dissolution upon ZnO uptake promotes the MAPK-dependent pro-inflammatory signaling pathway within immune cells.

The increase in Zn^2+^ release from ZnO NPs is correlated with the generation of ROS [70]. ZnO particles intrinsically produce low level of ROS, but production of ROS is significantly elevated by intracellular ZnO particles as the intracellular release of Zn^2+^ increases in the cytosol. Although the exact mechanism of the casualty of Zn^2+^ dissociation in ROS generation is unclear, previous studies suggested that extra Zn^2+^ ions displace key metal cofactors in metalloenzymes and metalloproteins within mitochondria and, thereby, damage mitochondrial membrane and increase intracellular superoxides [71]. Since ROS can modulate inflammatory signaling pathways, potentially via MAPK phosphatases inactivation [72], elevated intracellular ROS induces pro-inflammatory cytokines such as IL-6 by regulating transcriptional activity. Additionally, when ROS accumulates within cells, the NLRP3 inflammasome pathway is also activated [72] and active caspase-1 cleaves pro-IL-1β to induce the secretion of a pro-inflammatory cytokine, IL-1β.

### 2.4. Induction of Proinflammatory Cytokines and Costimulatory Molecules in Antigen-Presenting Cells

Immuno-modulating cascades induced by ZnO nanocomposites eventually converge into the production of pro-inflammatory cytokines. Previous studies have shown that ZnO NPs induce various inflammatory cytokines, such as IL-2, IL-4, IL-5, IL-6, IL-8, IL-17, IFN-γ, and TNF-α in phagocytic immune cells in vitro and in vivo [7,26,59,73]. Moreover, ZnO NPs also enhance the production of chemokines, such as CXCL-5, CXCL-9, and CXCL-10, which play a significant role in the trafficking of various leukocytes [74]. Interestingly, among several inorganic NPs (TiO_2_, silica, single-walled carbon nanotubes, multi-walled carbon nanotubes, and ZnO), ZnO NPs possessed the most drastic immunological effects, leading to higher expression of IL-1β and CXCL-9 in various antigen-presenting cells (APCs) [74]. One study showed a decrease in the chemotactic response of peripheral blood leukocytes to SDP-1α (also known as CXCL-12), a homeostatic chemokine that binds to its receptor CXCR4 and regulates hematopoietic cellular trafficking and secondary lymphoid organ architecture [75]. However, since the study was performed ex vivo, this effect of ZnO NPs needs to be further confirmed in vivo. 

In our previous studies, we consistently observed the stimulation of immunomodulatory and inflammatory responses in APCs, including macrophages and DCs, by ZnO-based nanocomposites [7,26]. ZnO NWs coated on PLLA microfiber significantly induced inflammatory cytokines, such as IL-1β, IL-6, IL-10, and TNF-α in DC2.4 cells, a dendritic cell line [26]. The surface expression of co-stimulatory molecules, including CD40 and CD86, significantly increased, whereas MHC class II molecules and CD80 marginally increased after stimulation with the ZnO-based nanocomposites. Enhanced expression of CD80 and CD86, as well as inflammatory cytokine IL-6 and TNF-α, in DCs upon exposure to ZnO NPs was consistently reported in a previous study [76].

## 3. Toxicity of ZnO Nanocomposites In Vivo

Increasing use of nanomaterials for medical application have raised concerns regarding their in vivo toxicity [77]. Comparative in vitro analysis of cellular toxicity using a human lung epithelial line, A549, exposed to various metal oxide particles (CuO, TiO_2_, ZnO, CuZnFe_2_O_4_, Fe_3_O_4_, and Fe_2_O_3_) and carbon NPs has been also well documented in a previous study [78]. Since many articles on the toxicity of ZnO nanocomposites are available [60,79,80,81,82,83,84], we briefly summarize current issues on its in vivo toxicity. As mentioned above, the potential toxicity of ZnO nanocomposites is generally attributed to the dissolution of the metal oxide complex into Zn^2+^ ions, especially in acidic environments [21]. Around 90% of ZnO NP mass dissolved within 24 h by incubation with artificial lysosomal fluid at pH 4.5, whilst ZnO NP in artificial interstitial fluid (Gamble’s solution, pH 7.4) showed no dissolution [85]. Therefore, the cytotoxic effect of ZnO NPs within cells and tissues is likely mediated by its ionic form rather than the particulate form [67]. Cellular uptake of ZnO NPs may facilitate cytoskeleton disturbances [86] and increases intracellular Zn^2+^ concentration, leading to rapid disruption of cellular Zn^2+^ homeostasis [87]. Consistently, inhibition of free Zn^2+^ ions in solution with EDTA abrogates ZnO NP-induced cellular toxicity in immune cells; increasing Zn^2+^ ion concentration by adding ZnCl_2_ induces similar toxicity [88]. Elevated Zn^2+^ ions results in intracellular ROS generation, DNA damage, and apoptotic cell death [84,89]. Addition of antioxidants, such as *N*-acetylcysteine or Trolox, abolishes inflammatory responses and DNA damage caused by ZnO NPs [90,91], suggesting a critical role of ROS generation. The degree of cytotoxic effect exerted by ZnO generally correlates with the release of Zn^2+^ and production of intracellular ROS and varies with the dose, their size, their shape, and surface modifications [60,84,92]. ZnO NPs at concentrations up to 100 μg/mL barely affect the viability of primary human peripheral blood mononuclear cells (PBMCs), whereas monocyte-derived dendritic cells (MDDC) cell death increased in a dose-dependent manner over 10 μg/mL of ZnO NPs [93]. Interestingly, ZnO NPs exhibit enhanced ability to kill cancerous cells by 28 to 35 times compared to normal cells [64]. Cellular apoptosis induced by ZnO NPs is reduced by inhibition of ROS in cancer cells [64], suggesting that oxidative stress induced by ZnO NPs may play a critical role in cellular apoptosis. It is also notable that smaller particles showed greater cellular toxicity, and surface modification of ZnO NPs using biocompatible polymers, such as PEG, Triton X-100, hyaluronan, and starch, reduces the toxicity of ZnO particles on cells [27]. Moreover, interactions with various in vivo proteins alter the cytotoxic effect of ZnO NPs through their surface properties [67]. Therefore, physicochemical characteristics of ZnO NPs and their dose ranges need to be finely adjusted to minimize cellular toxicity and enhance intended bio-functionality when they are applied in vivo.

### 3.1. In Vivo Toxicity by Systemically Administered ZnO Nanocomposites

Previous studies have reported that administration of ZnO nanocomposites could induce in vivo toxicity when systemically applied, i.e., intravenously or intraperitoneally [18,94]. Intravenous injection with ZnO nanorods in mice (2.4 mg/kg), significantly reduced red blood cells and platelet counts, elevated liver enzymes in sera and oxidative stress markers in the liver, and induced DNA damage in the liver, spleen, and kidney [94]. Intravenous injection of ZnO NPs in rats (30 mg/kg) also significantly increased mitotic figures in hepatocytes and induced multifocal acute injuries in lungs [20]. Another study compared intraperitoneal injection of ZnO NPs of different sizes (50 nm vs. 100 nm) and showed systemic zinc accumulation in the heart, liver, spleen, lung and kidney, especially with the smaller ZnO NPs [18]. Various in vivo toxicities observed in animal models systemically administered with ZnO nanocomposites have been well summarized in several recent reviews [59,60,89]. In general, high doses of systemically administered ZnO nanocomposites (more than ~mgs/kg of weight) induce systemic toxicity in various organs, especially in the liver, lungs, and kidneys, depending on the doses and sizes of the NPs. A few recent studies also reported neuronal toxicity in vivo after systemic administration [95]. Intraperitoneal injection of ZnO NPs (4 mg/kg body weight) twice weekly for 8 weeks in rats altered synaptic plasticity, which changed spatial learning and memory [96]. ZnO NPs (5.6 mg/kg) intraperitoneally administered three times per week for four weeks also induced production of pro-inflammatory cytokines in the serum and the brains of mice [97]. Repeated ZnO NP exposure in mice increased oxidative stress in the brain, impaired learning and memory abilities, and induced hippocampal pathological changes, especially in old mice. However, another study showed that repeated intravenous injection (five times at weekly intervals) of ZnO NPs (10 mg/animal) did not induce significant neuronal cell death or compromise the blood brain barrier in five week old rats [98]. In addition, systemic exposure of ZnO NPs (~μg/kg) in animal models did not significantly alter weight nor show organ toxicity. A comparative analysis with copper NPs with similar dose ranges revealed that ZnO NPs have less or no toxic effect in vivo [98], suggesting that ZnO NP doses below mg/kg are optimal and could have possible biomedical applications even when systemically administered. Nevertheless, the potential hazards of high concentrations of nanoscale ZnO on the central nervous system need further investigation.

### 3.2. In Vivo Toxicity by Locally Administered ZnO Nanocomposites

It has been shown that exposure of the respiratory tract to ZnO NPs can induce acute lung inflammation and systemic toxicity [92]. Oropharyngeal aspiration of ZnO NPs at 100 or 500 μg/kg in mice resulted in eosinophilic airway inflammation [61]. Significant neutrophilia on day 1 and significant eosinophilia in bronchoalveolar lavage fluid (BALF) at 7 days after exposure to high-dose (500 μg/kg) ZnO NPs were observed. The expression levels of cytokines IL-4, IL-5, and IL-13 significantly increased after 24 h of exposure to ZnO NPs and then gradually decreased. However, these inflammatory responses were not significant in mice exposed to low dose (100 μg/kg) of ZnO NPs, indicating that these are dose-dependent reactions. Intratracheal instillation of ZnO NPs (0.2 and 1.0 mg/rat) showed transient increase of neutrophil counts, expression of inflammatory chemokines, and oxidative stress markers in BALF, but these inflammatory responses generally disappeared within a week or a month [99]. In a human inhalation study, ZnO NPs was inhaled by healthy subjects with a concentration of 500 μg/m^3^ for 2 h, but found to be below the threshold for acute systemic effects as assessed by symptoms, leukocyte surface markers, hemostasis, and cardiac electrophysiology for up to 24 h post-exposure [100]. Nevertheless, high doses (over 1 mg/kg) or repeated exposure of ZnO NPs could be associated with dose-dependent mortality and severe pulmonary toxicity in rodent models [101]. Therefore, the optimal dose range of ZnO NPs needs to be carefully assessed when delivered via respiratory tract.

Subcutaneous administration of ZnO nanocomposites may have limited toxicity in vivo. In our previous study [19], subcutaneous injection of ZnO-coated iron oxide NPs up to 200 mg/kg in mice at weekly intervals for 4 weeks did not significantly induce systemic toxicity as measured by mortality, clinical observations, body weight, food intake, water consumption, urinalysis, hematology, serum biochemistry, and organ weights. A dose-dependent increase in granulomatous inflammation was observed at the injection site, but no other histopathological lesions in other organs could be attributed to the injected NPs. Zn concentration was not significantly higher in the sampled tissues, urine, or feces, but was increased at the injection site in a dose-dependent manner, along with a macroscopic deposition of the NPs. The presence of ZnO-coated iron oxide NPs at the injection site resulted in macrophage infiltration, but otherwise did not result in any systemic distribution or toxicity up to 200 mg/kg. We also observed a similar degree of local granulomatous inflammation at the injection site of radially grown ZnO NWs on PLLA microfibers [26] without significant systemic toxicity in mice injection model (data not published). Therefore, local subcutaneous injection of ZnO nanocomposites may not cause significant systemic toxicity, although further study is needed to investigate the local clearance of the nanocomposites and resolution process of foreign body reactions at the injection site.

### 3.3. Selective Toxicity to Cancer Cells by ZnO Nanocomposites

Many studies have reported that ZnO nanocomposites show selective cytotoxicity toward cancer cells in vitro and in vivo [102,103,104]. Hanley et al. reported that ZnO nanocomposites show preferential cytotoxic effect on cancer cells about 28–35 times more than normal cells [64]. Even though the exact mechanism of the selectivity is still unclear, physicochemical properties of ZnO NPs may confer cancer-specific killing effect. The isoelectric pH of ZnO NPs is 9–10, so ZnO NPs exhibit positive charge (ZnOH^2+^) under physiological conditions (~pH 7) [65]. The high metabolic demand of cancer cells leads to an accumulation of H^+^ ions in the tumor microenvironment, making gradients of acidosis [105]. Moreover, cancer cells frequently contain a high concentration of anionic phospholipids on their outer membrane and large membrane potentials. Interactions with positively charged ZnO nanoparticles are expected to be driven by electrostatic interactions, thereby promoting cellular uptake, phagocytosis, and ultimate cytotoxicity [104]. The increase of Zn ion concentration in the acidic environment leads to Zn-dependent protein activity disequilibrium, as well as increase of ROS concentration, leading to cytotoxicity of cells through oxidative stress and subsequent apoptosis [102,104,106]. In a recent study, Hu Y. et al. also demonstrated that ZnO NP-elicited autophagy contributes to tumor cell killing by accelerating the intracellular dissolution of ZnO and ROS generation [107]. In particular, ZnO NPs could promote Atg5-regulated autophagy flux without impairing autophagosome–lysosome fusion. Significant free zinc ion release in lysosomal compartments and sequential ROS generation were also associated with tumor cytotoxicity. They also showed enhanced antitumor effect of ZnO NPs in vivo in animal models injected with 4T1 tumor cells. The antitumor therapeutic effect of a combinatorial administration with ZnO NPs and doxorubicin obviously outperformed that of ZnO NPs or free doxorubicin treatment alone at the same dose, suggesting that autophagy induction by ZnO reinforces cancer cell death by enhancing zinc ion release and ROS generation when combined with other anti-cancer drugs [107].

Selective targeting of ZnO NPs to cancer tissues could be also facilitated by enhanced permeation and retention effect (EPR) because of their small size and surface properties [102]. Tumor tissues generally lack properly developed blood and lymphatic vessels and cellular connections of tumor cells are weak due to poor expression of tight junction. NPs can thereby easily diffuse through the blood vessels within tumor bed, thus showing enhanced permeation selectivity toward tumor cells [102].

## 4. Application of ZnO Nanocomposites for Vaccines and Immunotherapy

In order to elicit an adequate immune response, a series of processes are necessary for inducing proper adaptive immunity against specific protein antigens. First, the antigen needs to be efficiently delivered into APCs, such as DCs, and optimally processed and presented to T cells. Different methods of antigen uptake and lysosomal or cytoplasmic processing/presentation via major histocompatibility complex (MHC) I or MHC class II pathways have been proposed to account for the ability of a DC to stimulate either helper T (T_H_) cell or cytotoxic T lymphocyte (CTL) responses. Second, generation of specific subtypes of T cell responses against an insult is tightly controlled by regulating the differentiation of naïve T cells into effector CD4 and CD8 T cells via interactions with APCs. DCs play a significant role as translators between innate and adaptive immunity by integrating signals derived from tissue infection or damage and providing multiple soluble and surface-bound signals that help to guide T cell differentiation in secondary lymphoid organs (Figure 3) [108]. Therefore, a good vaccine system needs to leverage the complex cascades of antigen delivery into the APCs, functional activation status of DCs, and specific differentiation of T cells into the intended direction. A number of nanotechnology-based vaccines varying in composition, size, shape, and surface properties that have been approved or candidates for human use is increasing [10,11,109]. The immunomodulatory nature of ZnO nanocomposites confers great adjuvant potential when integrated into the vaccine formulation [102]. However, challenges remain due to a lack of fundamental understanding regarding the in vivo behavior of ZnO NPs, which can operate as either a delivery system to enhance antigen processing and/or as an immune-stimulating adjuvant to activate or enhance immunity [7,10,26]. In addition, there have been only a few studies investigating the adjuvant potential of ZnO nanocomposites in various forms for vaccines or immunotherapeutic applications (Table 1). Here, we summarize recent advances in prophylactic or therapeutic application of ZnO nanocomposites as vaccine adjuvant systems.

### 4.1. ZnO Nanocomposites as an Antigen Delivery System

A critical step of antigen processing and presentation by APCs to CD4 and CD8 T cells via the MHC class I and II pathways is the delivery route of the antigen [110]. Even though multiple cellular pathways exist for processing exogenous or endogenous antigen for presentation, antigen processing through the endocytic and/or phagocytic trafficking systems generally link to MHC class II-mediated presentation pathways to CD4 T cells. On the other hand, antigens in the cytosolic compartment are processed in a proteasome-dependent manner and presented via MHC class I molecules to CD8 T cells [110,111]. Therefore, efficacy of antigen delivery to appropriate cellular compartments is critical to determining the types and strength of antigen-specific T cell responses. Previously, our group had shown that ZnO nanocomposites can deliver an antigen into both compartments, thereby inducing significant and simultaneous elevation of CD4 and CD8 T cells specific to ZnO-associated antigens [7,8,26]. Antigens fused with ZBPs could not only be internalized through phagocytosis, but are also efficiently delivered into cytosolic compartment in APCs, potentially via direct penetration of cellular membranes [7,8,48]. Efficient colocalization of peptide antigens associated ZnO-based NPs with lysosomal compartments were consistently observed in DCs within a few hours of incubation [8]. It was shown that hollow ZnO nanospheres loaded with ovalbumin (OVA) dramatically enhance the intracellular uptake of the model antigen by APCs in vitro [24]. Robust intracellular delivery of associated peptides via direct membrane penetration was also confirmed by incubating HEK293 cells with unique fan-shaped ZnO NWs at 4 ℃. Since cellular activity, including endocytosis, is halted at 4 ℃, the presence of peptides coated on ZnO NWs within HEK293 cells suggested direct penetration of the associated peptides through cellular membrane [48]. Moreover, this 3D-structured ZnO nanocomposite could mediate the intracellular delivery of associated DNAs and result in their gene expression, clearly indicating that ZnO NWs transiently penetrate membranes to mediate intranuclear delivery of exogenous DNA [48]. Intracellular delivery of plasmid DNA and its gene expression by a ZnO tetrapods was also reported in other studies [46]. Considering that ZnO nanocomposites actively induce cellular autophagy [112], a major intracellular degradation system that delivers cytoplasmic constituents to lysosomes in both MHC class I and II-restricted antigen presentation, ZnO nanocomposites may therefore intrinsically further enhance intracellular antigen processing and/or presentation, although this needs to be verified in APCs. Collectively, ZnO-based nanocomposites possess promising characteristics in terms of antigen delivery into both endocytic/phagocytic compartments and cytosols, thereby simultaneously enhancing antigen processing and presentation via MHC class I and II pathways.

### 4.2. Induction of Antigen-Specific Adaptive Immunity by ZnO Nanocomposites

Upon antigen presentation by APCs to T cells in secondary lymphoid organs, activated CD4 T cells are differentiated into various subtypes of T cells, including T_H_1, T_H_2, T _H_17, and T_FH_ helper cells, with each supporting a distinct type of adaptive immunity [108]. Lineage-specific differentiation depends on the cytokine milieu of the microenvironment, as well as on the concentration of antigens, type of APC, and costimulatory molecules [113]. Effector mechanisms of each CD4 T cell subtype is primarily exerted by the secretion of different combinations of cytokines: (i) T_H_1 cells secrete IFN-γ, essential for the activation of monocytes and macrophages, and IL-2 promoting CD8 T cell proliferation and function, (ii) T_H_2 cells produce IL-4, IL-5, and IL-13 for coordinating humoral immunity and allergic inflammatory responses, (iii) T_H_17 cells generate IL-17 and IL-22 and mediate inflammatory responses and exhibit tissue protective properties, (iv) T_FH_ cells express IL-21 and costimulatory molecules to support humoral immunity through the interaction with B cells [113]. Previous studies revealed that ZnO-based nanocomposites can support antibody production and differentiation of CD4 T cells in various directions when co-administer with a specific antigen [7,8,25,50,73,114]. A study using 21 nm-sized ZnO NPs mixed with OVA antigen in suspension showed significant enhancement of T_H_2 response specific to OVA antigen when DBA/IJ mice were intraperitoneally immunized [73]. Antibody and T cell responses were biased to T_H_2 type as measured by enhanced production of OVA-specific IgG1 and IgE antibodies in sera, as well as elevated secretion of IL-4 and IL-5 from splenocytes in an OVA-specific manner, whereas T_H_1 responses as assessed by the levels of IgG2a and IFN-γ were relatively weaker. In addition, ZnO NPs had a stimulatory effect on the secretion of IL-17 in immune-splenocytes upon OVA stimulation [73], suggesting a shift toward production of T_H_17 subtypes. However, our previous studies using various forms of ZnO-based nanocomposites coated with high-affinity ZBP-fused protein antigens revealed efficient generation of antigen-specific T_H_1 responses, as well as CD8 T cells, in C57BL/6 mice immunized with antigen-loaded DCs or direct subcutaneous injection of the antigen complexes [7,8,26]. In addition, a significant increase of antigen-specific cellular immunity, as assessed by CTL activity, enhanced levels of IFN-γ-producing CD4 and CD8 T cells, and/or increased IgG2/IgG1 ratio, has been observed in various mice models immunized with antigen-absorbed ZnO tetrapod nanoparticles [50], mesoporous ZnO nanocapsules [25], and Zn-doped mesoporous silica NPs [114]. Therefore, ZnO-based nanocomposites might potently enhance T_H_1 responses and cell-mediated immunity against the associated antigen, depending on the efficacy of complex formation with the nanocomposites, adjuvant formulation, immunization route, and/or genetic background of host. Further studies on these issues need to be conducted in order to anticipate how to induce optimal T cell and antibody responses against a specific target insult.

### 4.3. Generation of Protective Immunity against Infections and Cancers by ZnO Nanocomposites

Only a few studies have confirmed the protective effect of ZnO-based nanocomposites as vaccine adjuvant for prophylactic or therapeutic applications in vivo against infections and cancers [7,8,9,24,26,50,114]. Dr. Shukla D.’s group have shown that specially designed ZnO tetrapod NP (ZOTEN) is an effective suppressor of HSV-2 genital infection in female BALB/c mice when used intravaginally as a microbicide [50]. The strong HSV-2 trapping ability of ZOTEN significantly reduced the clinical signs of vaginal infection and effectively decreased animal mortality. In parallel, ZOTEN promoted the presentation of bound HSV-2 virions to mucosal APCs, enhancing T cell- and Ab-mediated responses to the infection, and ultimately suppressing reinfection. Recently, they also showed that prior incubation of HSV-2 with ZOTEN inhibits the ability of the virus to infect vaginal tissues in female BALB/c mice and blocks virus shedding. Interestingly, the ZOTEN-neutralized virions elicit a local immune response that is highly comparable with HSV-2 infection, but with less inflammation and clinical manifestations [9]. This ZnO-based nanostructure directly binds to the virions, suppresses their infectivity, and facilitates the intracellular delivery of viral antigen into APCs, thereby enhancing antigen-specific adaptive immunity. This ZnO-based nanostructure utilizes the intrinsic anti-microbial properties of ZnO [115], as well as the immune modulating potential of the nanomaterial. Consistently, we also showed that a unique 3D design of ZnO NWs can dramatically enhance intracellular delivery of associated antigen into host cells [48]; immunization of a protein antigen, ScaA, complexed with ZnO NPs potently enhanced cell-mediated immunity against an intracellular pathogen, Orientia tsutsugamushi, providing protective immunity against the lethal pathogen challenge in vivo [7]. Therefore, the anti-microbial activity and immune modulating capacity of ZnO-based nanocomposites enables the expansion of their applicability as a novel vaccine platform to other infectious diseases.

In addition to acting as a vaccine adjuvant to protect against infections, ZnO-based nanocomposites can also significantly suppress tumor growth in vivo by enhancing anti-cancer immunity when combined with tumor-specific antigens. Wang X. et al. recently showed that prophylactic immunization of hollow ZnO NPs loaded with a model antigen, OVA, or autologous tumor antigen together with poly-inosinic:poly-cytidylic acid (poly(I:C)) inhibits cancer growth and metastasis to inguinal lymph nodes in a cancer cell challenge model [24]. They also found that pre-vaccination with mesoporous silica NPs doped with Zn significantly delayed tumor growth in vivo, potentially via inducing T_H1_ anticancer immunity [114]. In addition, we showed that mice immunized with DCs containing ZnO-coated iron oxide core–shell NPs had enhanced tumor antigen specific T-cell responses and therapeutically delayed tumor growth [8]. Moreover, a hybrid nanocomposite composed of ZnO NWs on PLLA fibers could successfully inhibit tumor growth when immunized with a tumor antigen even one week after tumor injection [26]. This ZnO-based nanocomposite also systemically reduced immune suppressive regulatory T cells and enhanced the infiltration of T cells into tumor tissues, compared to mice immunized with PLLA fibers coated with the antigen. These findings support the therapeutic potential of ZnO-based nanocomposites for treatment of established cancers.

### 4.4. Remaining Challenges

Despite a promising potential of ZnO nanocomposites as immune modulator, several challenges need to be resolved for the use ZnO nanocomposites as off-the-shelf adjuvant-like materials. First, mass production of homogenous ZnO nanocomposites with desired properties for therapeutic applications is the key challenge. ZnO nanocomposites hardly retain their stability and generally form aggregates in physiological aqueous system. The stability of ZnO nanocomposites could be improved by surface modifications with PEG, chitosan, and other polymers or organic/inorganic substances. In other words, it is inevitable that ZnO nanocomposite requires extra steps of surface modification to be fully functionalized. Interestingly, the stability of ZnO NPs in aqueous solution could be alleviated by poly(I:C) conjugation via electrostatic interactions between the positively charged ZnO NPs and the negatively charged dsRNAs. Proper combination of inorganic adjuvant (ZnO) and organic adjuvant (poly(I:C)) may further enhance therapeutic outcomes in vaccine and immunotherapy system [116]. Moreover, various physicochemical characteristics of synthesized ZnO nanocomposites, such as sizes, 3D shapes, compositions, and surface modifications, might differentially affect the interaction with in vitro and in vivo biological systems. Therefore, a cornerstone rule of thumb of the nano-bio interaction mechanism of ZnO needs to be specifically addressed. Additionally, many studies introduce different methods of facet-controlled synthesis of ZnO nanocomposites but sometimes they simply focus on fabricating different shapes and sizes of the material. Instead, investigation on the facet-specific characteristics of ZnO nanocomposites in term of nano-bio interaction would further advance our understanding for biomedical applications.

## 5. Conclusions

In this review, we compiled recent observations showing that ZnO nanocomposites could be applied as a potent immune adjuvant system with the proper composition, size, and 3D structure. Due to its peculiar physical and chemical characteristics, ZnO nanocomposites are readily taken up by APCs, dissolved to release Zn ions intracellularly, and subsequently induce generation of intracellular ROS. Nanocomposites are recognized by TLRs, and thereby efficiently induce activation of innate immune cells and promote pro-inflammatory responses. Activation of APCs, especially DCs, and delivery of associated antigen into intracellular compartments by ZnO nanocomposites enhance antigen-specific adaptive immune responses, such as antibody production and T cell responses. Even though the types of adaptive immunity potentiated by ZnO nanocomposites may vary depending on the injection route and intrinsic characteristics of the nanocomposite complexed with the antigen, optimal types of humoral and cellular immunity could be adjusted by engineering the physicochemical properties and 3D structures of the ZnO nanocomposites for specific target infections and cancers. Recent in vivo studies using ZnO nanocomposites with unique compositions and structures have provided promising proof-of-concept evidence for the development of prophylactic and/or therapeutic vaccines against infections and cancers. Nevertheless, the application of ZnO nanocomposites as vaccine delivery systems, as well as immunotherapeutic agents, is still at an early stage of development. Several challenges, including difficulties in reproducible mass production of ZnO nanocomposites with uniform characteristics and desirable properties, a lack of fundamental understanding of the interactions of nano-bio interfaces, and potential toxicity and bio-distribution of the nanocomposites during in vivo applications, needs to be resolved in future studies.

## Figures and Tables

**Figure 1 pharmaceutics-11-00493-f001:**
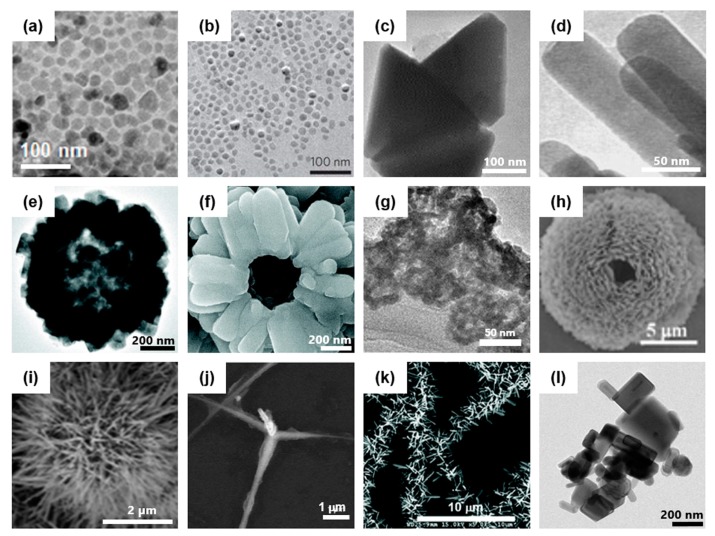
Artificially-synthesized ZnO-based nanocomposites with various sizes and shapes. (**a**) Transmission electron microscopic (TEM) image of ZnO NPs. Reproduced from Ha et al. [7] which is licensed under a Creative Commons Attribution-(CC BY 4.0) International License (http://creativecommons.org/licenses/by/4.0/) (**b**) TEM image of Fe_3_O_4_-ZnO coreshell nanoparticles (NPs). Reproduced with permission from Nature [8]; [2011], Springer. (**c**) TEM image of ZnO prism nanocrystals. Reproduced with permission from [52]; [2007], Elsevier. (**d**) TEM image of ZnO nanorods. Reproduced with permission from [53]; [2015], Elsevier. (**e**) TEM image of a hollow sphere particle composed of ZnO nanorods. (**f**) Scanning electron microscopic (SEM) image of typical particle (**e**). Reproduced with permission from [54]; [2008], American Chemical Society. (**g**) TEM image of hollow ZnO nanospheres used in vivo. Reproduced with permission from [24]; [2017], John Wiley and Sons. (**h**) SEM image of mesoporous ZnO hollow microsphere. Reproduced with permission from [55]; [2012], Royal Society of Chemistry. (**i**) SEM image of spiky particle composed of ZnO nanowires (NWs) grown on SiO_2_ nanoparticle. Reproduced with permission from [33]; [2015], Royal Society of Chemistry. (**j**) SEM image of tetrapod-like ZnO nanostructure. Reproduced with permission from [46]; [2006], John Wiley and Sons. (**k**) SEM image of radially grown ZnO NWs on poly(l-lactic acid) (PLLA) fibers. Reproduced from Sharma et al. [26] which is licensed under a Creative Commons Attribution-(CC BY 4.0) International License (https://creativecommons.org/licenses/by-nc/3.0/). (**l**) TEM image of commercially available ZnO ‘Calamine’ powder.

**Figure 2 pharmaceutics-11-00493-f002:**
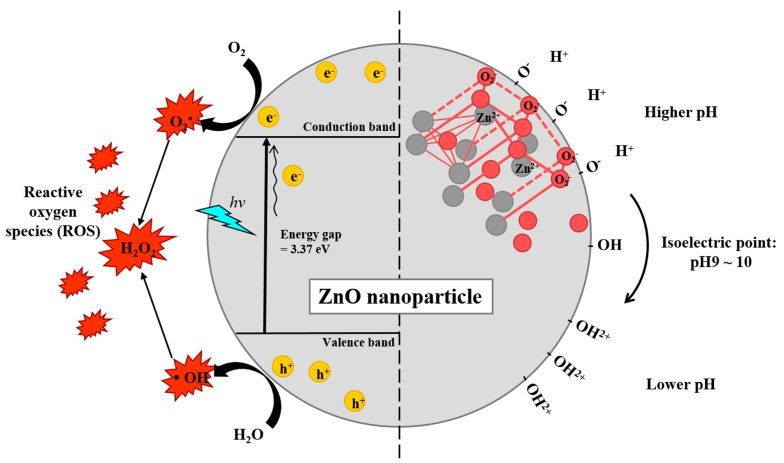
Schematic diagrams of potential mechanisms generating reactive oxygen species (ROS) by ZnO nanocomposite (left) and surface charge shift depending on environmental pH (right). ZnO has a large bandgap energy (3.37 eV) separated by the conduction band and valence band. When the ZnO absorbs light in UV range, electrons (e^−^) get excited and promoted into the conduction band from the valence band, leaving holes (h^+^). Both excited electrons and holes are strong agents in ROS generation. The crystallite structure of ZnO is usually hexagonal wurtzite structure in which each anion is surrounded by four cations at the corners of a tetrahedron, and vice versa, respectively. ZnO has an isoelectric point at pH 9–10. The neutral hydroxyl groups on their surface either gain or lose protons from and into the surrounding, depending on the surrounding pH. Therefore, ZnO nanocomposites carry positive surface charge under physiological conditions (pH < 7.4), aiding their intracellular uptake.

**Figure 3 pharmaceutics-11-00493-f003:**
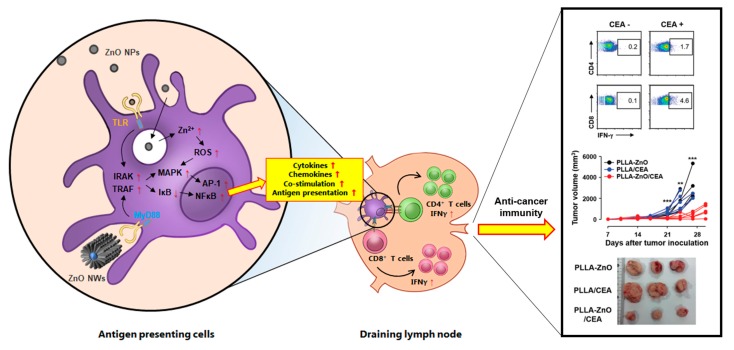
Schematic diagram of the proposed effects of ZnO nanocomposites on antigen-presenting cells and their potential role in inducing T cell responses in draining lymph node. Various forms of ZnO nanocomposites (e.g., ZnO NPs and NWs) can stimulate inflammatory responses in antigen-presenting cells via TLR recognition or direct endocytosis/phagocytosis, which subsequently initiates immune signaling cascades. TLR stimulation by ZnO nanocomposites can upregulate downstream molecules IRAK and TRAF, and enhance subsequent MAPK pathways. Then, the inhibitor of NFκB, IκB, is degraded and NFκB is activated accordingly. Endocytosed ZnO nanocomposites transported into the acidic lysosomes dissociate into Zn^2+^ ions and ionic form of zinc disrupts oxidative balance, increasing ROS generation. Ultimately, upregulation of pro-inflammatory signaling pathway and ROS generation enhance the production of pro-inflammatory cytokines and chemokines, and increase the surface expression of co-stimulatory and antigen presenting molecules. Stimulated antigen presenting cells migrate into draining lymph node and prime naïve CD4 and CD8 T cells into effector T cells, which can initiate both humoral and cell-mediated immune responses against specific antigen associated with ZnO nanocomposites. A representative example of anti-cancer immunity induced by ZnO nanocomposites coated with a tumor antigen, carcinoembryonic antigen (CEA), is presented in the right panel (Reproduced from Sharma et al. [26] which is licensed under a Creative Commons Attribution-(CC BY 4.0) International License (https://creativecommons.org/licenses/by-nc/3.0/).). NP: nanoparticle, NW: nanowire, TLR: Toll-like receptor, MyD88: myeloid differentiation primary response protein-88, IRAK: IL-1 receptor associated kinase, TRAF: TNFR-associated factor, MAPK: mitogen-activated protein kinase pathway, NFκB: nuclear factor κ chain enhancer of activated B cells, IκB: inhibitor of κB, AP-1: activator protein 1, ROS: reactive oxygen species.

**Table 1 pharmaceutics-11-00493-t001:** Application of ZnO nanocomposites in development of vaccines and cancer immunotherapy in vivo.

ZnO Nanocomposites	Disease/Antigen *	Host Mouse	Vaccination Route	Biological Responses **	Reference
ZnO NP	N.A./OVA	DBA/1J	Intraperitoneal	Increase in IL-4, IL-5, and IL-17; increase in IgG1 and IgE	[73]
Fe_3_O_4_-ZnO coreshell NP	Cancer/CEA	C57BL/6	Subcutaneous (DCs)	Enhanced IFN-γ^+^ CD4 and CD8 T cells; delayed tumor growth	[8]
ZnO NP	N.A./OVA	BALB/c	Intraperitoneal	Increased inflammation in intestine	[22]
ZnO tetrapod (ZOTEN)	HSV2/HSV2	BALB/c	Intravaginal	Enhance T cell and Ab responses; decreased mortality	[50]
ZnO NP	Scrub typhus/ScaA	C57BL/6	Subcutaneous	Enhanced IFN-γ^+^ CD4 and CD8 T cells; protective immunity	[7]
ZnO NP/poly(I:C)	Cancer/N.A.	BALB/c	Intratumoral	Suppress tumor growth	[116]
Hollow ZnO NP	Cancer/aTA	C57BL/6J	Subcutaneous	Enhanced CD4 and CD8 T cells; delayed tumor growth	[114]
Mesophorous ZnO NP	N.A./OVA	BALB/c	Subcutaneous	Enhanced IFN-γ^+^ CD4 and CD8 T cells, elevated IgG2	[25]
ZnO NWs on PLLA fiber	Cancer/CEA	C57BL/6	Subcutaneous	Enhanced IFN-γ^+^ CD4 and CD8 T cells, delayed tumor growth	[26]
ZnO tetrapod (ZOTEN)	HSV2/HSV2	BALB/c	Intravaginal	Blocks viral shedding and reduced inflammation	[9]

*, N.A.: not associated, TA: tumor antigens, aTA: autologous tumor antigens; ** Ab: antibody.

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
