# Peer review of "Application of ZnO-Based Nanocomposites for Vaccines and Cancer Immunotherapy"

_pharmaceutics, 2019, doi:10.3390/pharmaceutics11100493_

Round 1
Reviewer 1 Report
The inorganic nanocomposites for vaccine formulation represent interesting compounds and the research about them is promising.
The review gives an overview of the recent literature about the use of the ZnO nanocomposites as vehicle for therapeutic vaccines against tumors and infection, underlying their toxicity. In my opinion the paragraph 3 enlisting the toxicities is detailed but its length do not help the reader. Moreover a list of the benefit when compared to less toxic compounds should be added. Moreover it will be interesting mention the possibility to translate the compound in a off the shelf drug, once overcome the toxicity hindrance
I've enlisted below some points in need of clarification
Lane 72: "Dioscorides introduces ZnO in the first century"
It is not clear what the ZnO is introduced for
lane 118: "ZnO particles are readily ingested"
Please add the ZnO particles dimension
Figure 2 add a more detailed description of the left and right side of the figure, what conduction and valence band means and how the energy gap is filled?
Figure 3 the caption does not explain the mechanism, please elaborate and add an abbreviation description
lane 227 "when applied in vivo applications"
it seems a word is lacking
lane 171 "The increase in Zn2+ release from ZnO NPs is correlated with the generation of ROS. "
The ROS generation is mentioned several times in the text, but a description of how it happens is lacking
lane 310 "vessels towards a tumor cells" means vessels towards a tumor cell?
lane 344 ZBP should be explained
Author Response
Responses to reviewer #1
Comments #1-1: The inorganic nanocomposites for vaccine formulation represent interesting compounds and the research about them is promising.
The review gives an overview of the recent literature about the use of the ZnO nanocomposites as vehicle for therapeutic vaccines against tumors and infection, underlying their toxicity. In my opinion the paragraph 3 enlisting the toxicities is detailed but its length do not help the reader. Moreover a list of the benefit when compared to less toxic compounds should be added. Moreover it will be interesting mention the possibility to translate the compound in a off the shelf drug, once overcome the toxicity hindrance
Response #1-1: We are grateful for the reviewer’s positive feedback and valuable comments.
We agree with the reviewer that an additional comparison between ZnO and less toxic materials would be nice and straightforward. FDA acknowledges bulk ZnO as generally recognized as safe (GRAS) substance and ZnO NPs are typically biocompatible. Although ZnO is generally considered to be a material with low toxicity, ZNO is not completely inert and sometimes more toxic than other inorganic materials. However, ZnO may possess several advantages over those materials: i) ZnO is more than just a drug carrier and exerts an adjuvant potential itself, modulating inflammatory responses. ii) By 24h of co-incubation of ZnO with normal cells such as monocytes and macrophages in vitro, ZnO NPs are fully dissociated into Zn2+ ions, leaving no particulate forms in the culture. This could possibly minimize adverse effects such as vascular congestion that would otherwise occur with particle accumulation in the body. iii) ZnO is multifunctional that its unique physicochemical properties are being exploited in many fields such as cosmetics, bio-imaging, food packaging, water purifications, and tother. iv) Low cost fabrication, various structure, and facile surface modification of ZnO are also important advantages over many other inorganic NPs. Comparative explanations on the inflammatory activity and cellular toxicity of ZnO NPs were presented in the revised manuscript (line 211 ~ 214, 235 ~ 238, 254 ~ 256).
We agree with your comments and off-the-shelf drug of ZnO nanocomposites would be wondrous but challenges remain besides the toxicity of the material. We addressed about such challenges and possible future studies required to make ZnO more therapeutically approachable as a future drug in a new section (Line 484 – 500).
Lastly, there are other review papers available that introduce the toxicity of ZnO nanocomposites so we summarized the very key factors that are noteworthy in the current review, mainly stressing on the in vivo toxicities of ZnO nanocomposites in the previous studies. Hope our response to this particular comment is satisfactory and we highly appreciate the reviewer’s comment.
Comments #1-2: Lane 72: "Dioscorides introduces ZnO in the first century" It is not clear what the ZnO is introduced for
Response #1-2: We agree that the particular information was vague. It is presumed that ZnO ointment was introduced for topical applications of eye and skin by Dioscorides. We revised the manuscript accordingly in line 91. Dioscorides had studied many inorganic metal and ZnO is one of them. It is interesting that ZnO has been investigated for therapeutic use since such long ago.
Comments #1-3: Lane 118: "ZnO particles are readily ingested" Please add the ZnO particles dimension
Response #1-3: We agree with the reviewer in that the information was rather missing. Sahu et al. has confirmed that ZnO NPs up to 5 μm in size showed little difference in the phagocytic capability of human monocytes. The manuscript was revised in line 147 as suggested.
Comments #1-4: Figure 2 add a more detailed description of the left and right side of the figure, what conduction and valence band means and how the energy gap is filled?
Response #1-4: Thanks for your advice and we have now edited the requested information in the figure caption (line 172 – 179). Conduction band means the electron orbitals of free electrons and valence band means the outermost orbital that electrons actually occupy. The bandgap energy is the energy difference between the highest occupied energy state of the valence band and the lowest unoccupied state of the conduction band. A large bandgap energy means that a lot of energy is required to excite electrons into the conduction band. Normally metals have an overlap in between two bands so the electrons get readily jump between the two bands. Semiconductors such as ZnO does not have this overlap in bands and is separated by the bandgap. When an external energy in UV ray range is applied, electrons in the valence band gets excited and promoted into the conduction band, leaving holes (h+) in the valence band. According to the reviewer/s comments, we revised the manuscript for clarification.
Comments #1-5: Figure 3 the caption does not explain the mechanism, please elaborate and add an abbreviation description
Response #1-5: We thank for reviewer’s advice and revised the manuscript in figure caption (line 195 – 209) including the abbreviation descriptions.
Comments #1-6: Lane 227 "when applied in vivo applications" it seems a word is lacking
Response #1-6: It was corrected as recommended (line 281).
Comments #1-7: Lane 171 "The increase in Zn2+ release from ZnO NPs is correlated with the generation of ROS.” The ROS generation is mentioned several times in the text, but a description of how it happens is lacking
Response #1-7: Indeed, this is a really important point we missed in the original submission, and we thank the reviewer for pointing this out. However, the exact mechanism of the correlation between Zn2+ dissolution and the subsequent ROS generation is still unclear and most of the past studies presume the correlation relying on the results that more ROS was detected with greater intracellular Zn2+ presence. Yet, some theories include that extra Zn2+ ion displace the core cofactors of mitochondria which results in the disruption of mitochondrial membrane. This subsequently disrupts the oxidative balance of a cell, increasing superoxide within the cell and generating ROS. We revised the manuscript, including the information in line 222 – 225.
Comments #1-8: lane 310 "vessels towards a tumor cells" means vessels towards a tumor cell?
Response #1-8: We agree that the sentence was rather misleading and have edited as follows in line 363: “NPs can thereby easily diffuse through the blood vessels within tumor bed, thus …”
Comments #1-9: lane 344 ZBP should be explained
Response #1-9: As mentioned in the manuscript line 123, ZBP is an abbreviation of ‘ZnO binding peptide’ which means a peptide that binds to ZnO NPs specifically with high affinity.
Reviewer 2 Report
The authors Sharma et al. present an interesting and complete overview about the application of ZnO nanoparticles for vaccines and cancer immunotherapy. The authors show along the review appropriate examples of using ZnO nanoparticles for that purpose. In general, the review is logical and well written. Otherwise, I may purpose some minor changes.
- Line 49: FDA abbreviation should be explained. I suggest to the authors to add an abbreviation list at the end of the review. There are several abbreviations and information may be lost.
- Line 76. The authors wrote: “ZnO is a semiconductor material with a direct wide bandgap energy (3.37 eV) and a large excitation binding energy (60 meV) at room temperature”. What is the consequence/further application of that? Comparison with other materials could be appropriated.
- In figure caption of Figure 1, check carefully the use of the abbreviation NP or NPs. Please use NP when there is only one nanoparticle and abbreviation NPs when there are many.
- In line 136 it says “Of note, the majority of ZnO particles (~ μm in diameters)…” Is it the μm or nm? If the size of the ZnO particles is in micrometers the name of the composites along the manuscript must be changed to microparticles/micro composites. If the work is focused in both (micro and nanoparticles), the authors must write micro/nano-particles (or similar). This correction include the tittle.
- Line 350: The authors should specify the kind of cells.
- Line 351: there is an author comment.
- I suggest to the authors to include a new figure to ilustrate one application (at least) of the ZnO nanoparticles for vaccines and immunotherapy (in vivo or/and in vitro).
Author Response
Responses to reviewer #2
Comments #2-1: The authors Sharma et al. present an interesting and complete overview about the application of ZnO nanoparticles for vaccines and cancer immunotherapy. The authors show along the review appropriate examples of using ZnO nanoparticles for that purpose. In general, the review is logical and well written. Otherwise, I may purpose some minor changes.
Response #2-1: We thank very much for the Reviewer’s interest and valuable comments.
Comments #2-2: Line 49: FDA abbreviation should be explained. I suggest to the authors to add an abbreviation list at the end of the review. There are several abbreviations and information may be lost.
Response #2-2: Thanks for your advice. We agree with the reviewer and edited abbreviations in the text where needed (line 47 ~ 48, 206 ~ 214).
Comments #2-3: Line 76. The authors wrote: “ZnO is a semiconductor material with a direct wide bandgap energy (3.37 eV) and a large excitation binding energy (60 meV) at room temperature”. What is the consequence/further application of that? Comparison with other materials could be appropriated.
Response #2-3: It is a really nice point that we should address. We revised the manuscript and the appropriate applications of the semiconductor property of ZnO are now described in line 96 – 107.
Comments #2-4: In figure caption of Figure 1, check carefully the use of the abbreviation NP or NPs. Please use NP when there is only one nanoparticle and abbreviation NPs when there are many.
Response #2-4: Thanks for your advice. We edited the abbreviations and sentences where needed in line 134 and 139.
Comments #2-5: In line 136 it says “Of note, the majority of ZnO particles (~ μm in diameters)…” Is it the μm or nm? If the size of the ZnO particles is in micrometers the name of the composites along the manuscript must be changed to microparticles/micro composites. If the work is focused in both (micro and nanoparticles), the authors must write micro/nano-particles (or similar). This correction include the tittle.
Response #2-5: We agree with the reviewer that the paragraph might be misleading and it was meant to be micron. The explanation was to briefly mention how different sizes of ZnO particles can be recognized in different ways and we thought it would be an interesting note for the readers. Yet our scheme was focused on nano-sized ZnO composites, especially in vivo applications of ZnO nanocomposites. Hope our response to this particular comment is satisfactory and we highly appreciate the reviewer’s comment.
Comments #2-6: Line 350: The authors should specify the kind of cells.
Response #2-6: Thanks for the advice and the cells were HEK293. We edited the information in the manuscript in line 404.
Comments #2-7: Line 351: there is an author comment.
Response #2-7: Thanks for the advice and we have deleted the comment and provided addition explanation for the comment (line 405 ~ 407).
Comments #2-8: I suggest to the authors to include a new figure to illustrate one application (at least) of the ZnO nanoparticles for vaccines and immunotherapy (in vivo or/and in vitro).
Response #2-8: Great point and we now have added an additional figure in Figure 3 (line 195 ~ 209) about the representative in vivo application of ZnO nanocomposite in cancer.
Reviewer 3 Report
In this review, the authors provide insights into ZnO-based nanocomposite for applications of vaccines and cancer immunotherapy. They summarized the history of ZnO applications in biomedical fields and then more focused on vaccines and immunotherapy. The review, however, did not give a comprehensive summary of the development of the biomedical application of ZnO-based nanocomposites. I do not think it could be published on Pharmaceutics at this moment. Some comments as follows need to be addressed.
The introduction section is too short and general. The authors summarized the application history of the ZnO particles in the biomedical field, which could give an overview of this topic. However, the challenges of the topic should be clearly emphasized. Ways to address these challenges or potential solutions should be included.
Need a section to summarize the methods to make ZnO particles with different shapes. How the size and shape of the ZnO particles influence their biomedical applications, especially in the focused application areas reviewed in this paper?
The authors often use an expression like “as mention above,” “see below,” which is pretty vague to refer the information the authors would like to share. Please revise it and be specific.
In line 350 and 351, It is unclear what blocks the cellular endocytosis at 4 oC. A clear statement and references are needed.
Author Response
Responses to reviewer #3
Comments #3-1: In this review, the authors provide insights into ZnO-based nanocomposite for applications of vaccines and cancer immunotherapy. They summarized the history of ZnO applications in biomedical fields and then more focused on vaccines and immunotherapy. The review, however, did not give a comprehensive summary of the development of the biomedical application of ZnO-based nanocomposites. I do not think it could be published on Pharmaceutics at this moment. Some comments as follows need to be addressed.
Response #3-1: We are very grateful for reviewer’s valuable comments and revised the manuscript with the reviewer’s advices as follows.
Comments #3-2: The introduction section is too short and general. The authors summarized the application history of the ZnO particles in the biomedical field, which could give an overview of this topic. However, the challenges of the topic should be clearly emphasized. Ways to address these challenges or potential solutions should be included.
Response #3-2: We thank for reviewer’s advice and added the appropriate further studies and challenges for ZnO nanocomposites in order to be used in vivo applications in a separate section 4.4 (line 483 – 500).
Comments #3-3: Need a section to summarize the methods to make ZnO particles with different shapes. How the size and shape of the ZnO particles influence their biomedical applications, especially in the focused application areas reviewed in this paper?
Response #3-3: We appreciate very much of the reviewer’s valuable comments. The scope of our review was more focused on the in vivo application of ZnO nanocomposites in vaccines and immunotherapy and originally excluded the synthesis methods of the material. Yet, we agree with the reviewer and found the need to address briefly on the ZnO synthesis methods and therefore added an additional section (Section 1.1 (line 63 – 82)), explaining current strategies in ZnO nanoparticle synthesis.
Comments #3-4: The authors often use an expression like “as mention above,” “see below,” which is pretty vague to refer the information the authors would like to share. Please revise it and be specific.
Response #3-4: Thanks for pointing this out and we agree with the reviewer. We revised the manuscript where such expressions were made in line 231 ~ 232, 341.
Comments #3-5: In line 350 and 351, It is unclear what blocks the cellular endocytosis at 4 ℃. A clear statement and references are needed.
Response #3-5: We agree with the reviewer that the sentence was rather misleading. When cells are incubated in 4℃, cellular activity is halted. Therefore, cells cannot perform endocytosis and the intracellular presence of peptides coated on ZnO NWs must have been delivered through direct penetration. We revised the manuscript with the information in line 403 - 407.
Round 2
Reviewer 3 Report
Authors answered all my comments. I think it could be accepted for publication.